# Visualising trends in dentition to lip mouth morphology using geometric morphometrics

**Tobias M. R. Houlton** [1]*, **Nicolene Jooste**[2], **Maryna Steyn**[3], **Jason Hemingway**[3]

**1** Centre for Anatomy and Human Identification, School of Science and Engineering, University of Dundee, Dundee, United Kingdom, **2** Department of Human Anatomy and Physiology, University of Johannesburg, Doornfontein, Johannesburg, South Africa, **3** Human Variation and Identification Research Unit (HVIRU), School of Anatomical Sciences, University of the Witwatersrand, Medical School, Parktown, Johannesburg, South Africa

\* thoulton001@dundee.ac.uk

**Data Availability Statement:** All relevant data are within the manuscript, with raw data provided in the Supporting Information files.

**Funding:** TMRH was awarded the Study Abroad Studentship by the Leverhulme Trust (UK), grant

## Abstract

Linear measurements taken from bony landmarks are often utilised in facial approximation (FA) to estimate and plan the placement of overlying soft tissue features. This process similarly guides craniofacial superimposition (CFS) practices. Knowledge of how hard and soft tissue features spatially relate around the mouth region is, however, limited. Geometric morphometric techniques have thus been used to investigate size and shape variation in dentition-to-lip mouth morphology in a South African population. Twenty landmarks (twelve dentition, eight lips) were digitised, using cone-beam CT images of the anterior craniofacial complex in a Frankfurt/Frankfort position, for 147 individuals aged between 20 and 75 years. Principal Component Analysis and Canonical Variate Analysis established that much shape variation exists. A two-way ANOVA identified significant (p < 0.0001) population and sex variation with mouth shape. Black individuals presented with thicker lips, with the oral fissure aligning closely to the dental occlusion. Oral fissure position for white individuals corresponded to the inferior one-quarter (females) or one-sixth (males) of the maxillary central incisor crowns. Males presented larger dimensions than females, but females had a greater lip-to-teeth height ratio than their male counterparts. A pooled within-group regression analysis assessed the effect of age on the dentition and lips and found that it had a significant (p < 0.0001) impact on mouth shape. Ageing was associated with a reduced lip and teeth height, increased mouth width, and a lowered oral fissure and cheilion placement. The generated mean shape data, with metric guides, offer a visual and numerical guide that builds on existing FA and CFS standards, enhancing our understanding of hard and soft tissue relationships.

## Introduction

Craniofacial identification practices assist human identification cases via the analysis and/or generation of facial images. Techniques such as facial approximation (FA) and craniofacial superimposition (CFS) offer indispensable methods of police intelligence in otherwise

number: SAS-2017-005 (URL: https://www.leverhulme.ac.uk/study-abroad-studentships). The funders had no role in study design, data collection and analysis, decision to publish, or preparation of the manuscript.

**Competing interests:** The authors have declared that no competing interests exist.

unsolvable cases. They play a particularly critical role in South Africa. The South African Police Service (SAPS) often utilises FA and CFS in a medico-legal environment overwhelmed by unidentified decedents. In 2020, the Gauteng Health Department reported that 1,173 unidentified decedents had entered the forensic pathology services in Gauteng (the most populous province of South Africa) [1]. Several factors influence this statistic [2]. South Africa has a high mortality rate from violent crime and poor road safety. Identification of deceased individuals is hindered by considerable movement within the country by South African nationals, and both documented and undocumented immigrants. In all instances, there can be poor or irregular communication with friends and family, and inadequate identification documentation. Regular forms of forensic identification, such as dental records and DNA profiling, are also limited. Dental records are rare, due to the poor socio-economic status of many residents and lack of oral health facilities and dental practicioners in the country [3]. Fingerprints may be unavailable, and such analyses are limited to documented residents and criminal convicts. No DNA database exists. FA and CFS thus form a crucial last resort for tracing a decedent's identity.

FA and CFS depend on an anatomical understanding of how soft and hard tissues of the face relate. Traditionally, facial characteristics have been studied based on linear distances and proportion indices [4]. Although FA and CFS guidelines using these data offer a fundamental insight into soft tissue estimation from the skull, a clearer guide to how soft tissue structures spatially relate to the skull is essential. Geometric morphometrics is grounded on a mathematical theory of shape that captures considerable spatial information encoded in landmark coordinates. This approach is being increasingly used in face shape analysis, with existing studies investigating facial development and variation [5–8], facial masculinity [9], and comparing facial shape between monozygotic and dizygotic twins [10]. No existing geometric morphometric studies, however, has considered the spatial relationship between hard and soft facial tissues of the mouth in a South African population.

The mouth is an example where some information is available that estimates height and width dimensions, but the exact shape of the lip vermillion, however, remains to be one of the most error-prone areas in FA [11]. Existing orthodontic and anatomical literature indicate that orolabial morphology is influenced, amongst other things, by the occlusion of the teeth, dental pattern, and facial profile [12]. The position of the oral fissure has often been indicated to be at one-third or one-quarter of the maxillary incisor crown height superior to the dental occlusal line [12–15]. Other reports suggest that the oral fissure bisects the mid-line of the maxillary incisor crowns [16] or the occlusive line of the teeth [17]. The corners of the mouth are often described as being positioned on radiating lines from the first premolar-canine junction [12,18–20], or should be calculated so that the inter-canine distance comprises 75% of the overall mouth width [21]. Other sources state that the mouth corners should be positioned inferior to the infraorbital foramina [22]. Sets of regression formulae have also been developed to assist in lip height and mouth width estimation [20,23,24]. Much of the aforementioned guidelines were obtained from cadaver data, or from faces that were recorded in supine positions. These obviously have many shortcomings, as gravity as well as postmortem dehydration and distortion negatively impacts the reliability of the findings. Even though metric information of the hard and soft tissue relationship is thus available, deducing the shape of the mouth from the features of the skull is particularly problematic. The shortage of information regarding the shape of the mouth results in considerable artistic interpretation. This is concerning because the mouth and lips play a key role in the evaluation and recognition of the craniofacial complex [12].

This study aimed to evaluate patterns in lip vermillion shape and position in relation to the underlying dentition, using geometric morphometric analysis on serial craniofacial cone-

beam computerised tomography (CBCT) scans. It utilised an adult South African sample, which predominantly represented black individuals (Sub-Saharan African ancestry), and a small white sample (European ancestry; limited by scan availability). The shape patterns identified are intended for use in conjunction with pre-existing regression equations [e.g., 23,24].

## Materials and methods

This study utilised 147 CBCT scans of South African adult patients. This was a retrospective study that utilised patient data held by the University of Pretoria, Oral and Dental Hospital. Patients had given informed written consent for their data to be used in scientific research, under the ethical governance of the University of Pretoria. Permission to access data was therefore passed by the institutional research ethics committee (clearance number: 212/2016), and all procedures performed in this study were in accordance with the 1964 Declaration of Helsinki and its later amendments or comparable ethical standards.

A Planmeca Promax 3D Max scanner was used, with images taken at 200μm voxel size, 96kV, and 10mAs. Individuals were scanned in an upright position. Patients were aged between 20 and 75 years. Population origins were self-prescribed. The sample comprised of forty-one black females (mean age 37.1 years, SD 12.1 years), sixty-seven black males (mean age 32.7 years, SD 9.0 years), twenty white females (mean age 35.5 years, SD 12.8 years), and nineteen white males (mean age 36.7 years, SD 11.2 years). Subjects were selected on the basis that they demonstrated a neutral facial expression, with no distinct evidence of intrusive craniofacial trauma, congenital anomalies, extensive tooth loss, or surgery impeding the basic craniofacial appearance, especially of the mouth.

CBCT scans were initially viewed and prepared using the 3D DICOM viewer, OsiriX. Hard and soft tissue anterior 2D stills were captured in JPEG format, with the craniofacial complex orientated on the Frankfurt/Frankfort Horizontal Plane (FHP). Positioning was achieved using both porions and both orbitale landmarks. Four reference planes were created using the different combinations of three landmarks and an average of those four planes defined a mean horizontal plane. Landmark digitisation was performed using ImageJ version 1.50i [25]. Twelve hard tissue and eight soft tissue landmarks were assigned, with corresponding coordinates collected for each individual (Table 1, Fig 1). Although hard and soft tissue landmarks were collected using independent images for each individual, the identical setting of each image enabled the data sets to be integrated for geometric morphometric analysis. To assess repeatability, one trained investigator repeated landmark allocation three times on all the available scans (intra-observer). The landmark allocation was repeated after an interval of at least two weeks. Additionally, 20% of the total sample (n = 30) underwent repeat landmark allocation by another trained investigator (inter-observer). To evaluate repeatability, a Procrustes ANOVA was performed [26,27].

Geometric morphometric techniques [28] captured size and shape variation from the mouth and dentition coordinate data. All morphometric analyses were carried out using MorphoJ version 2.0 [29]. Shape information was extracted with a generalized full Procrustes fit projected to shape tangent space [30]. We considered only the symmetric component of shape variation as asymmetry was not of interest for this particular study. Centroid size was computed as a measure of mouth and dentition size. To examine the effect of age on mouth morphology, a pooled within-group regression was performed that accounted for differences between sex and among population groups. The significance of the regression was assessed against the null hypothesis of independence by randomly re-associating shapes and sizes among individuals 10,000 times [31]. After accounting for age, a Principal Component Analysis (PCA) was used to examine the dominant features and dimensionality of shape variation

**Table 1. Assigned landmarks for geometric morphometric analysis.**

**Hard tissue landmarks**

| Fig 1(a) ref. | Landmark | Definition |
|---|---|---|
| 1 | Maxillary lateral canine (right) | The most lateral point, midway down the length of the right maxillary canine |
| 2 | Maxillary central-lateral incisor junction (right) | A point on the alveolar border midway between the right maxillary central incisor and maxillary lateral incisor |
| 3 | Superior midpoint of maxillary central incisor (right) | A midpoint at the cementoenamel junction of the right maxillary central incisor |
| 4 | Prosthion | A point on the alveolar arch midway between the central maxillary incisors |
| 5 | Superior midpoint of maxillary central incisor (left) | A midpoint at the cementoenamel junction of the left maxillary central incisor |
| 6 | Maxillary central-lateral incisor junction (left) | A point on the alveolar border midway between the left maxillary central incisor and maxillary lateral incisor |
| 7 | Maxillary lateral canine (left) | The most lateral point, midway down the length of the left maxillary canine |
| 8 | Cusp of maxillary canine (right) | A point on the cusp of the right maxillary canine |
| 9 | Incision (superior) | A point where the maxillary central incisors meet on the occlusal line. |
| 10 | Cusp of maxillary canine (left) | A point on the cusp of the left maxillary canine |
| 11 | Incision (inferior) | A point where the mandibular central incisors meet on the occlusal line. |
| 12 | Infradentale | The apex of the alveolar between the mandibular central incisors |

**Soft tissue landmarks**

| Fig 1(b) ref. | Landmark | Definition |
|---|---|---|
| 13 | Cheilion (right) | A point located at the outermost right corner (commissure) of the mouth where the upper and lower lips meet. It demarcates the lateral extent of the labial fissure |
| 14 | Peak of Cupid's bow (right) | A point on the border of the upper lip vermillion, at the right peak of the Cupid's bow |
| 15 | Trough of Cupid's bow | A point on the border of the upper lip vermillion, at the central dip of the Cupid's bow |
| 16 | Peak of Cupid's bow (left) | A point on the border of the upper lip vermillion, at the left peak of the Cupid's bow |
| 17 | Cheilion (left) | A point located at the outermost left corner (commissure) of the mouth where the upper and lower lips meet. It demarcates the lateral extent of the labial fissure |
| 18 | Stomion (superior) | A midpoint of the oral fissure marked on the upper lip margin |
| 19 | Stomion (inferior) | A midpoint of the oral fissure marked on the lower lip margin |
| 20 | Lower lip (labrale inferius) | A point where the boundary of the vermilion border of the lower lip and the skin is intersected by the median sagittal plane |

[30,32]. A Canonical Variate Analysis (CVA) was similarly performed, which is a discriminatory analysis that calculates the linear variation that best discriminates between multiple groups (i.e., black female, black male, white female, white male). Mean shape data according to each population and sex group was subsequently extracted for quantitative analysis. After reintroducing the mean size into the landmark data for each group, a series of inter-landmark distances were calculated using Pythagorean Theorem, which metrically demonstrates how the lip vermilion and dentition relate. Relevant inter-landmark distances that contextualise the relative lip and dentition measurements within and between groups were also collected (Table 2). Using PAST version 4.03 [33], a two-way ANOVA assessed the significance of population and

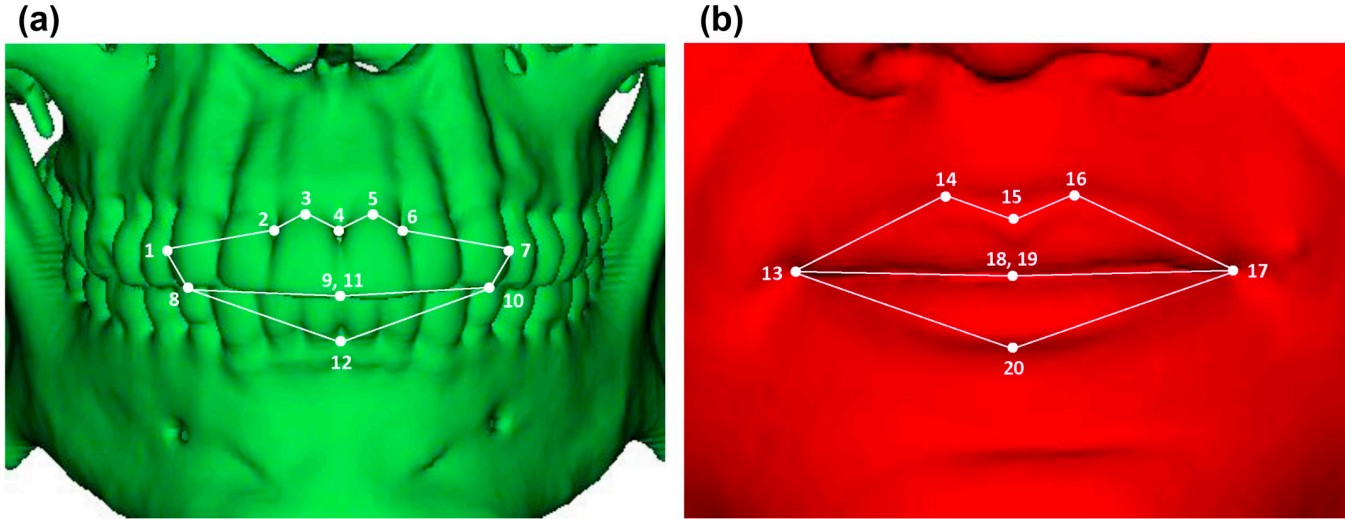

**Fig 1.** Visual reference to hard (a) and soft (b) tissue landmark coordinates collected for analysis (refer to Table 1 for landmark definitions).

**Table 2. Definitions of hard tissue, soft tissue, and relative hard to soft tissue measurements collected.** All measurements are taken at a right angle from the denoted landmarks.

| Hard tissue measurements | Definition | Landmarks (Table 1) |
|---|---|---|
| Total occluded central incisors height | Taken parallel to the long axis, it is the longest apicocoronal distance between the most apical point of the maxillary cementoenamel junction to the most apical point of the mandibular cementoenamel junction | 3–12 (y) |
| Maxillary central incisor height | Taken parallel to the long axis, it is the longest apicocoronal distance between the most apical point of the cementoenamel junction and most incisal point of the anatomical crown, of the maxillary central incisor | 5–9 (y) |
| Mandibular central incisor height | Taken parallel to the long axis, it is the longest apicocoronal distance between the most apical point of the cementoenamel junction and the visible incisal point of the anatomical crown, of the mandibular central incisor | 11–12 (y) |
| Inter-canine width | Measured between the most lateral borders of the maxillary canines | 1–7 (x) |
| **Soft tissue measurements** | **Definition** | **Landmarks (Table 1)** |
| Total lip height | Taken from the most superior and inferior points denoting the vermillion ridge/border of the upper lip and lower lip | 16–20 (y) |
| Upper lip height | Taken from the most superior point denoting the vermillion ridge/border of the upper lip, to the most anterior point of contact between the upper and lower lips | 16–18 (y) |
| Lower lip height | Taken from the most anterior point of contact between the upper and lower lips to the point denoting the vermillion ridge/border of the lower lip in the midsagittal plane | 19–20 (y) |
| Cupid's bow width | Measured between the two most superior peaks of the vermilion upper lip, which form the base of the philtral columns | 14–16 (x) |
| Mouth width | Measured between the lateral-most aspects of the angle of the mouth on each side | 13–17(x) |
| **Relative hard to soft tissue measurements** | **Definition** | **Landmarks (Table 1)** |
| Superior dentition to lip (from Cupid's bow) | Taken between the most superior aspect of the upper lip vermilion border of the Cupid's bow, to the most superior aspect of the maxillary central incisors cementoenamel junction | 5–16 (y) |
| Superior dentition to lip (medial) | Taken on the midsagittal plane between the central trough of the upper lip vermilion border, to the most superior aspect of the maxillary central incisors cementoenamel junction | 4–15 (y) |
| Inferior dentition to lip | Taken on the midsagittal plane between the lower lip vermilion border, to the most inferior aspect of the maxillary central incisors cementoenamel junction | 12–20 (y) |
| Lateral canine to cheilion | Taken from the most lateral point of the maxillary canine to the outermost corner of the mouth | 7–17 (x) |
| Incision to stomion | Taken on the midsagittal plane between the point of dental occlusion and lip occlusion | 9–18/19 (y) |

**Table 3. Digitisation error table of the Procrustes ANOVA.**

|  | Df | SS | MS | $R^2$ | F | P |
|---|---|---|---|---|---|---|
| Individual | 147 | 10.701 | 0.073 | 0.999 | 1520.8 | <0.0001 |
| Intraobserver | 296 | 0.014 | $4.8 \times 10^{-5}$ | 0.001 | | |
| Total | 443 | 10.715 | | | | |
|  | Df | SS | MS | Rsq | F | P |
| Individual | 29 | 2.467 | 0.085 | 1.000 | 8762.8 | <0.0001 |
| Interobserver | 30 | $2.9 \times 10^{-4}$ | $9.7 \times 10^{-6}$ | $1.2 \times 10^{-4}$ | | |
| Total | 59 | 2.467 | | | | |

sex, including their interaction, for each linear measure. The Holm-Bonferroni method was used to account for the raised type I error-rate resulting from having run the 14 tests.

## Results

A frequentist analysis identified that the mean Procrustes distance between repeated measurements (intra- and inter-observer; raw data are available under supporting information) fell below 95% of the Procrustes distances between individuals, indicating that measurement error contributed less than 5% to the observed shape variation (Table 3).

The within-group regression suggested that age was significantly associated with mouth shape (p < 0.0001) and accounted for ~14% of the within-group variance observed (predicted sums of squares: 0.293; total sums of squares: 2.032). Fig 2 illustrates results from the pooled within-group regression against age. The generated scatter plot indicates that age similarly impacts each tested population and sex group. The corresponding wireframe diagrams demonstrate that an increase in age can be associated with visible lip thinning and decrease in teeth height. The soft tissue mouth furthermore enters a more inferior placement relative to the dentition, mostly impacting the upper lip, oral fissure and cheilions placement. Mouth width also increased, with a slight broadening and reduced emphasis of the upper lip Cupid's bow.

After accounting for age, a general PCA was conducted. Common patterns in morphological variation are expressed in PC1 to PC4, which accounted for 84% of the total variance (PC1 = 39%, PC2 = 28%, PC3 = 10%, PC4 = 7%) (Figs 3 and 4). PC1 demonstrates visible

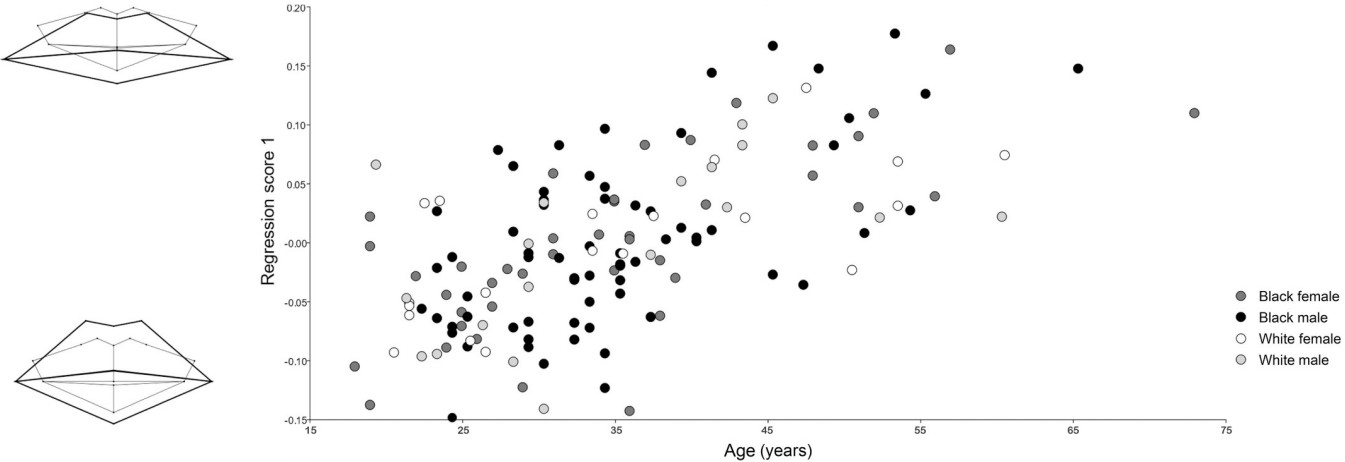

**Fig 2. Age regression results.** Wireframe models depict the greatest possible difference between lip shape (thick black outline) and dentition shape (thin grey outline) in younger (inferior model) and older (superior model) individuals.

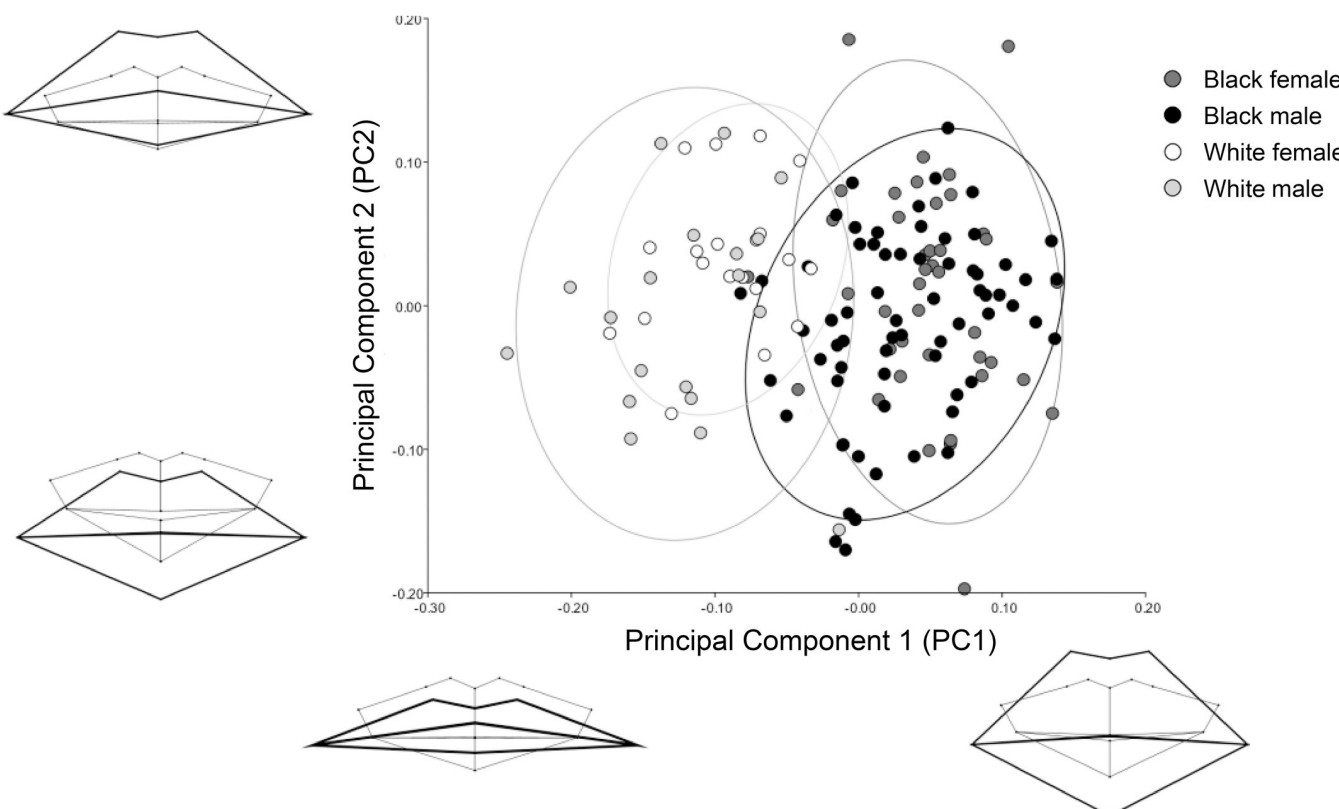

**Fig 3. Principal component (PC) analysis identifying the greatest possible variations in mouth morphology after accounting for age.** PC1 represents 39% and PC2 28% of the total variance. PC1 identifies clear population differences.

population differences. The South African white group often presented with a wide, thin-lipped mouth, with total lip height fitting within the dental margins, and oral fissure positioned approximately one-quarter of the maxillary incisor height superior to the dental occlusion. The South African black group often presented a narrower, thick-lipped mouth, where the total lip height exceeds the dental margins, and the oral fissure is approximately level to the dental occlusion. PC2 to PC4 appear to demonstrate shape variations that are unrelated to population or sex. PC2 illustrates that as the mandibular incisors increase in height, the soft tissue mouth shifts from a superior to inferior placement relative to the teeth, and the cheilions elevate to align closely with the stomion. PC3 and PC4 identify shifts in a slightly open to closed dental occlusion (likely influenced by how individuals' clench or relax their mouth), with the lower lip slightly thickening with a closed dental occlusion. This shape consideration is likely related to postural behaviour at individual level. In PC3, as the mouth widens, the oral fissure drops from a slight superior to inferior position relative to the dental occlusion, and the cheilions rise to horizontally align with the stomion.

CVA analysis indicated that CV1 and CV2 accounted for 74% and 15% of the between group variance, respectively (Fig 5). CV1 and PC1 present similar population specific morphological variances. CV2 alternatively demonstrates a shifting emphasis in the prominence of the central maxillary incisors. As the emphasis increases, the soft tissue mouth enters a superior placement relative to the dentition. Sexual dimorphism is evident in the white group, but less visible in the black group. Mahalanobis distances identified that black males and females demonstrated the most morphometric similarity ($D^2$ = 1.2323, p = 0.0015), whereas white females and black males were the most different ($D^2$ = 3.7254, p < 0.0001) (Table 4).

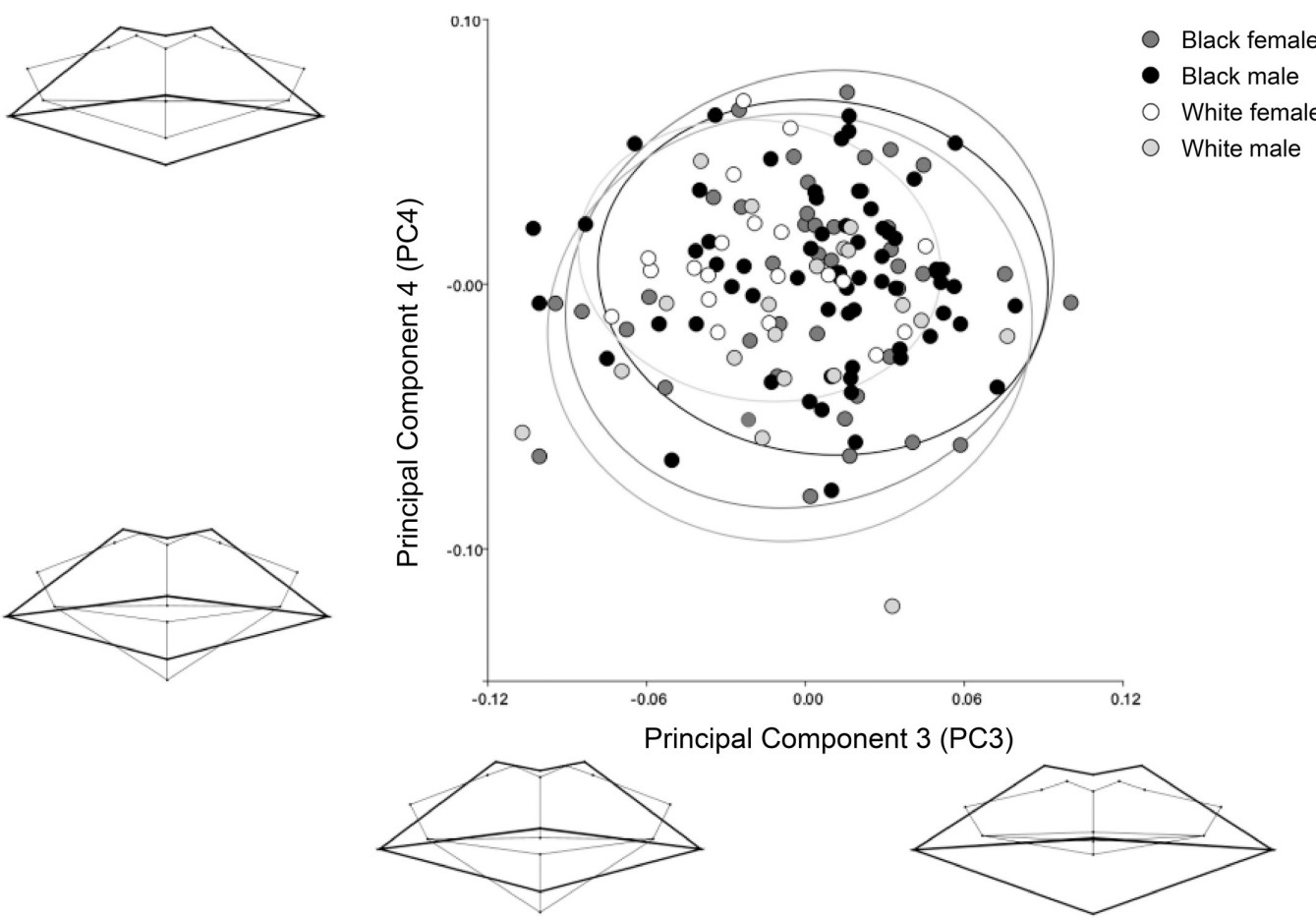

**Fig 4. Principal component (PC) analysis identifying the next greatest possible variations in mouth morphology following PC1 and PC2 results, after accounting for age.** PC3 represents 10% and PC4 7% of the total variance.

Mean mouth dimensions and geometric wireframe diagrams were generated for each population and sex group (Table 5, Fig 6). Significant population and sex differences were identified, but no significant sex differences existed between population groups. Sexual dimorphism thus presented in the same way for both populations.

Black individuals presented significantly (p< 0.0001) bigger dimensions compared to white individuals for inter-canine width, lip height, and Cupid's bow width. Dentition-to-lip measurements also identified that in black individual's the lips exceed the underlying dental margins, whereas white individual's lips tend to be within the dental margins. White individuals also presented with a greater difference between the dental occlusion and stomion of the lips compared to black individuals. When affiliating the measured mean maxillary central incisor height with the mean dental occlusion to stomion distance, the location of the stomion with relation to the teeth can be identified. The stomion was found to take a superior placement from the dental occlusion, bisecting the maxillary central incisor tooth height by 1/17 in black females, 1/37 in black males, 1/4 in white females, and 1/6 in white males.

Males were significantly (p < 0.0001) larger compared to females in total occluded central incisor height, maxillary central incisor height, inter-canine width, Cupid's bow width, and mouth width. Only black males presented with significantly (p = 0.04) larger upper and total lip heights. Females, however, tended to have a proportionally greater total lip height relative

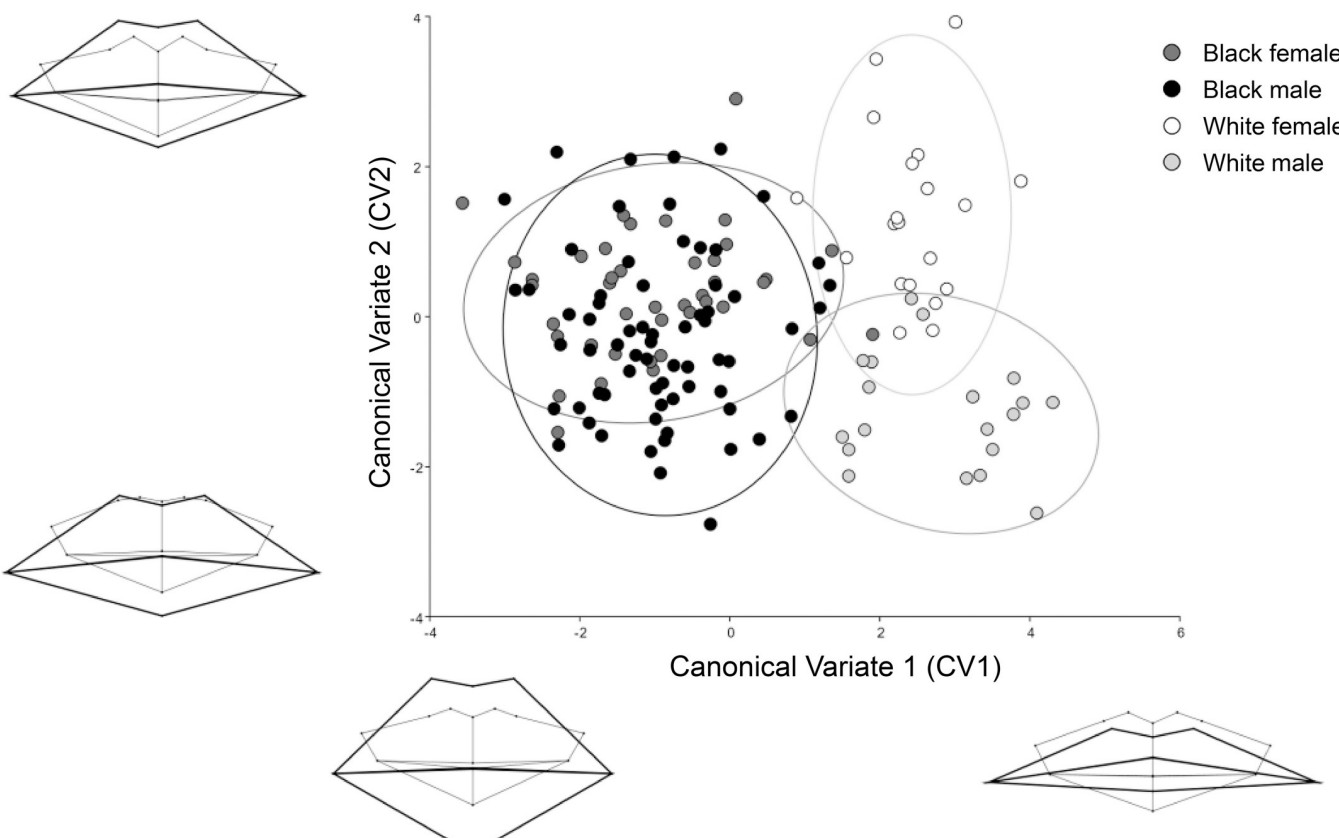

**Fig 5. Canonical variate (CV) analysis identifying the greatest possible variations in mouth morphology after accounting for age.** CV1 identifies clear population differences. CV2 identifies sexual dimorphism.

to teeth height compared to males. Calculated mean ratios for total central incisors' height to total lip height were 1:1.43 in black females, 1:1.41 in black males, 1:0.9 in white females, 1:0.82 in white males.

## Discussion

Locating soft and hard tissue landmarks on the face can be error prone [34]. A high agreement in intra- and inter-observer repeatability was, however, achieved in this study, suggesting a good level of reliability in the overall results. All data were recorded with the subject in an upright seated position, eliminating the postural changes associated with a supine position [35].

**Table 4. Mahalanobis distances among groups (P-values from 10,000 permutations).** Black female (BF), black male (BM), white female (WF), and white male (WM).

| Comparison | D$^2$ | P |
|---|---|---|
| BF-BM | 1.2323 | 0.0015 |
| WF-WM | 2.7425 | < .0001 |
| BF-WF | 3.7224 | < .0001 |
| BM-WM | 3.9585 | < .0001 |
| BF-WM | 4.1735 | < .0001 |
| WF-BM | 3.7254 | < .0001 |

**Table 5. Mean dental and lip measurements, with standard deviations in parentheses, including relative measurements between dental and lip landmarks, for each population and sex group.** Measurements are in millimetres.

**Hard tissue measurements**

| Population | Sex | n | Total occluded central incisors height | Maxillary central incisor height | Mandibular central incisor height | Inter-canine width |
|---|---|---|---|---|---|---|
| Black | Female | 41 | 17.5 (3.0) | 10.2 (1.0) | 6.4 (1.8) | 40.8 (2.4) |
| | Male | 67 | 18.6 (3.0) | 11.0 (1.0) | 6.9 (1.9) | 43.4 (2.2) |
| White | Female | 20 | 16.3 (1.9) | 10.5 (0.8) | 5.7 (1.7) | 38.8 (1.7) |
| | Male | 19 | 17.5 (3.0) | 10.7 (1.4) | 6.3 (2.0) | 41.4 (2.6) |
| P value differences | Population: | | 1 | 1 | 0.73 | $6.57 \times 10^{-6}$ |
| | Sex: | | $3.71 \times 10^{-5}$ | $5.37 \times 10^{-3}$ | 1 | $5.78 \times 10^{-10}$ |
| | Interaction: | | 0.07 | 1 | 1 | 1 |

**Soft tissue measurements**

| Population | Sex | n | Total lip height | Upper lip Height | Lower lip height | Cupid's bow width | Mouth width |
|---|---|---|---|---|---|---|---|
| Black | Female | 41 | 25.0 (4.1) | 12.5 (2.2) | 12.4 (2.3) | 14.6 (1.9) | 53.7 (4.6) |
| | Male | 67 | 26.3 (4.1) | 13.3 (2.3) | 12.9 (2.1) | 16.0 (2.0) | 56.2 (4.8) |
| White | Female | 20 | 14.6 (2.7) | 7.1 (1.5) | 7.4 (1.9) | 13.3 (1.7) | 51.2 (3.1) |
| | Male | 19 | 14.4 (3.4) | 6.9 (1.8) | 7.5 (2.1) | 15.0 (1.8) | 59.0 (5.5) |
| P value differences | Population: | | $2.82 \times 10^{-31}$ | $1.36 \times 10^{-29}$ | $6.51 \times 10^{-25}$ | $7.68 \times 10^{-3}$ | 1 |
| | Sex: | | 0.04 | 0.04 | 0.21 | $7.08 \times 10^{-5}$ | $3.71 \times 10^{-5}$ |
| | Interaction: | | 0.64 | 0.55 | 1 | 1 | 0.07 |

**Relative dental to lip measurements**

| Population | Sex | n | Superior dentition to lip (from Cupid's bow) | Superior dentition to lip (medial) | Inferior dentition to lip | Lateral canine to cheilion | Incision to stomion |
|---|---|---|---|---|---|---|---|
| Black | Female | 41 | 3.0 (2.8) | 3.2 (3.0) | -4.5 (3.4) | 6.4 (2.5) | 0.6 (2.6) |
| | Male | 67 | 2.6 (3.1) | 2.8 (3.4) | -5.1 (3.3) | 6.4 (2.1) | 0.3 (2.1) |
| White | Female | 20 | -0.4 (2.4) | 0.4 (2.4) | 1.3 (3.1) | 6.2 (1.4) | 2.9 (2.0) |
| | Male | 19 | -2.1 (3.0) | -2.3 (3.1) | 1.1 (2.3) | 8.8 (2.2) | 1.7 (2.5) |
| P value differences | Population: | | $7.04 \times 10^{-10}$ | $4.48 \times 10^{-8}$ | $4.10 \times 10^{-17}$ | 0.23 | $5.55 \times 10^{-4}$ |
| | Sex: | | 1 | 1 | 0.71 | 1 | 0.78 |
| | Interaction: | | 1 | 0.73 | 1 | 0.05 | 1 |

Using geometric morphometrics, it was found that age significantly (p < 0.0001) impacted mouth morphology, affecting all population and sex groups in a similar way (Fig 2). This study delivered a unique wireframe visual that demonstrates the impact of age on soft and hard tissue mouth shape, which is largely consistent with metric descriptions previously reported by other authors [14,18–20,23,24,36–38]. The identified age-related lip-thinning, reduction in teeth height, and increase in mouth width, are well recognised in published literature [14,18–20,36–38]. The current study additionally identified that with the decrease in lip height and increase in mouth width, the Cupid's bow similarly flattens and widens. The oral fissure and cheilions furthermore drop to a more inferior placement with age. This has been similarly reported in a small longitudinal cephalometric study by Akgül and Toygar [37], who found that during the third decade of life the lips become thinner and gravitate downward. Tentative reasons for these age-related changes in facial dimensions include microscopic factors (reduction in elastic fibres, skin elasticity and resilience; thinning of the cutis; muscle weakness and reduction; increase in subcutaneous fat) and macroscopic factors (gravity; changes in posture;

### Black Female
n = 41, mean age 37.1 years

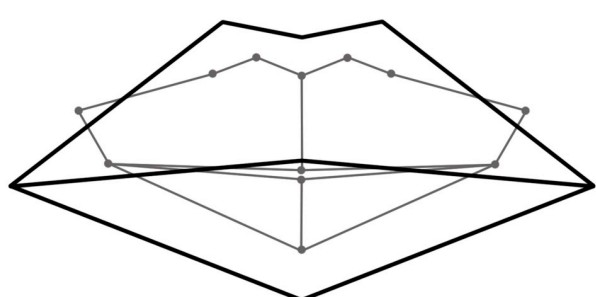

### Black Male
n = 67, mean age 32.7 years

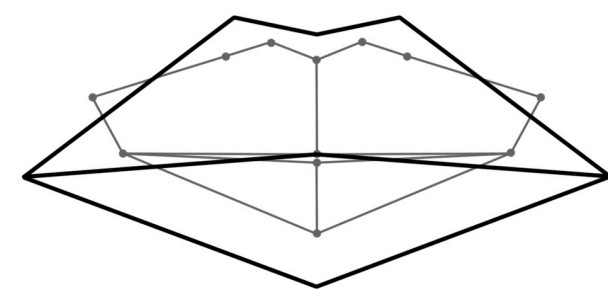

### White Female
n = 20, mean age 35.5 years

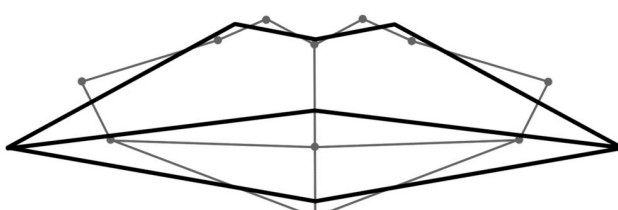

### White Male
n = 19, mean age 36.7 years

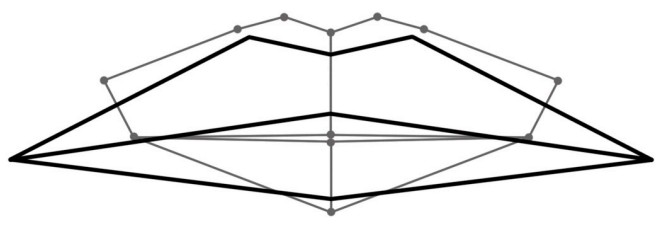

**Fig 6. Mean lip shape (thick black outline) and dentition shape (thin grey outline) variations according to each investigated population and sex group.**

weight gain or loss) [12,36]. Dental wear is otherwise influenced by attrition, abrasion, and erosion over time [12,18,19].

Generated wireframes delivered a first-time visualisation of lip-to-teeth shape patterns that present according to population (Figs 3 and 5, PC1 and CV1). Although minimal differences existed in tooth height metrics between populations, there was a significant (p < 0.0001) difference in lip height. On average, black individuals' total lip height exceeded the total central incisor crown height by 7.6 mm. The white group presented lips that aligned more closely within the cementoenamel junctions, presenting an average total lip height that was 2.4 mm less than the total height of the central incisor crowns. The popularly cited [12,15,18,39], European derived, lip height approximation guideline that indicates lip height to be approximately equal to the height of the cementoenamel junctions, is therefore comparable for white South Africans but does not apply to black South Africans. Evident population variation in lip heights have been similarly identified in other studies accessing black South African and white Italian participants [38], and European and Indian Sub-continent participants [20]. Black individuals also presented a significantly (p < 0.0001) greater inter-canine width and Cupid's bow width compared to white individuals. Unlike existing research [40,41], black individuals did not demonstrate significantly wider mouths compared to white individuals. This may be a result of sample bias, due to the limited white sample size available in this study. Other studies with a greater representation of white individuals (total n = 1223) [21,41–43] reported mean mouth width measurements ranging from 51.8–56.6 mm in white males, and 47.4–51.4 mm in white females. Compared to this study, the white female sample corresponds with the existing literature, but the white male sample exceeds it. Previously published mean mouth width

measurements for black males otherwise range from 55–60 mm (n = 216) [38,41], with black females only once being reported to have a mean mouth width of 52 mm (n = 28) [41]. These results more closely correspond to this study (males = 56.2 mm, SD 4.8; females = 53.7 mm, SD 4.6). All these findings consequently suggest that population specific regression formulae [i.e., 20,23,24], supported by the morphometric guides generated in this study, are currently needed to assist with mouth approximation.

Shape differences denoted in the presented wireframe diagrams illustrate population variance in cheilion placement (Figs 3 (PC1), 5 (CV1), and 6). In white individuals, the cheilions take on a more inferior placement relative to the stomion than with black individuals. White individuals consequently present a more laterally downturned oral fissure angle, while black individuals present a more neutral oral fissure angle. This is similarly identified in black male facial average images generated by Schmidlin et al. [38], where a minimal downturn of the mouth corners presents only after 60 years. In this study, however, age had less emphasis on creating a downturned oral fissure angle, but instead caused the overall lip structure to thin and droop into a more inferior placement with relation to the teeth (Fig 2). Unrelated to age and population, PC2 (Fig 3) related a neutral oral fissure angle with an increase in the visible mandibular incisor's height, which synchronised with the soft tissue mouth shifting from a superior to inferior placement relative to the teeth. Oral fissure angle could be influenced by the underlying muscle and skeletal structure, as hyper tensed depressor anguli oris muscles could depress the cheilions [44]. We also propose that the prognathic profile prevalent in black individuals may offer a structural support that helps elevate the modioli and maintain a horizontal oral fissure angle. The orthognathic profile common in white individuals, may alternatively offer less support and might allow the weight of the soft tissues from the cheeks to have a greater gravitational effect on the cheilions. White individuals also presented with a moderately thicker lower lip compared to upper lip dimensions. A fuller form of the lower lip might have elevated the stomion and emphasised the downturned angle of the cheilions. In all cases, the cheilions are positioned just inferior to the dental occlusion, except for white females, where they are level with the dental occlusion.

Fig 6 illustrates the mean dentition-to-lip landmark relationships, generated for each population and sex group. Oral fissure position for white individuals, especially white females, approximately agrees with past reports indicating that the opening rests on the inferior one-third or one-quarter of the maxillary central incisor crown height, superior to the dental occlusion [12–15]. On average, in white females and males, the stomion is 2.9 mm (2.0 SD) and 1.7 mm (2.5 mm) superior to the dental occlusion, respectively. Black individuals alternatively had an oral fissure that aligned closely with the dental occlusion. On average, black females and black males respectively presented the stomion 0.6 mm (SD 2.6) and 0.3 mm (SD 2.1) superior to the dental occlusion. The difference in oral fissure placement may be related to variances in facial profile and overall facial proportions, to allow appropriate spacing of lower face facial features over the skull, for the mechanical movement and expression of the mouth. Thicker lips, prevalent in black individuals, might also experience a slightly greater gravitational pull, causing the oral fissure to align more closely to the dental occlusion than with thinner lips. This feature may furthermore be impacted by the fact that the upper lip was on average thicker than the lower lip in black individuals.

Shape differences between the sexes were not as pronounced as population differences; only the results observed in CV2 indicate some degree of dimorphism between white males and females (Fig 5). This is consistent with existing publications that focus on measurements [14,36,40,41], however, males often presented larger hard and soft tissue dimensions than females (Table 5). This is likely due to a proportional difference, where males tend to demonstrate a larger craniofacial complex, rather than greater mouth morphology to face ratio.

Female lips do, however, tend to appear "fuller" in relation to the underlying dentition than those of males, which is evident in mean shape patterns identified in Fig 6. Using available mean measurements, females were found to present slightly greater lips-to-teeth height ratios than their male counterparts. This trend could be related to females having significantly smaller total teeth heights (p < 0.0001), rather than any specific difference in total lip thickness (p = 0.04), compared to males. In white females, the total occluded central incisor height was 1.2 mm smaller and total lip height 0.2 mm thicker than white males. The smaller dental height measurements thus emphasised the slightly thicker lip proportions. Black females presented a 1.1 mm smaller total occluded central incisors height and 1.3 mm smaller total lip height than black males. They still, however, presented with a marginally greater lip-to-teeth height ratio (1.43:1) than black males (1.41:1).

The mean wireframe templates generated (Fig 6), with reference to the PCA and CVA findings (Figs 3, 4 and 5), and metric guides that empirically indicate hard and soft tissue landmark relationships (Table 4), deliver novel shape data that is intended to substantiate and dynamically support existing linear approaches to mouth estimation [i.e., 20,21,23–24].

Age, population, and to a lesser extent, sex variables, were identified to impact the shape, size, and proportions of the mouth. Visual patterns establish that the points of the Cupid's bow are typically located within the margins of the maxillary central-lateral incisor junctions. The oral fissure closely bisects the dental occlusal line in black individuals but enters a more elevated position in white individuals. In white individuals, a general agreement with existing literature can be made, namely that the mouth opening is situated approximately one-quarter of the maxillary central incisor crown height superior to the dental occlusal line [12–15]. A downturned oral fissure was common in white individuals, while a more horizontal alignment was prevalent in black individuals. In the white group, the total soft tissue mouth is within the margins of the teeth, closely aligning with the cementoenamel junctions, whereas the opposite was true for black individuals. Males typically presented bigger hard and soft tissue dimensions compared to females, but females demonstrated a greater lip-to-teeth height ratio than their male counterparts. The performed regression on age supports the notion that an increase in age is characterised by a gradual elongation of the mouth and lip thinning.

The aim of this study was to evaluate morphometric patterns in lip vermillion shape and position in relation to the underlying dentition, to support existing mouth approximation guidelines used in FA and CFS practices [i.e., 20,21,23,24]. The mean wireframe templates generated that differentiate population and sex categories (Fig 6), with reference to the PCA and CVA findings (Figs 3 to 5), and metric guides (Table 5), have consequently been generated as a visual guide, for use in conjunction with existing mouth approximation formulae [i.e., 20,21,23,24].

This current study was limited by available scan data. Ideally, a greater and more evenly distributed sample size (with relation to population, sex, and age variables), should be developed to improve the geometric morphometric model and subsequent results. In future, a detailed investigation into the effects of allometry will furthermore support research developments in this area. This study, however, is the first of its kind to graphically demonstrate how hard and soft tissue features of the mouth relate, offering a visual and more holistic guide that surpasses the limits of existing approximation methods using metric data alone.

## Conclusion

From this study, we have largely identified shape, size and proportional differences related to demographic variables such as age, population, and sex. Other, more universal, variables that are unrelated to demographics were also identified (Figs 3 and 4, PC2 to PC4). Shape and size

changes related to dental prominence, dental occlusion, and mouth width were found. Some of this variation may be influenced by intrinsic structural differences in biological form, including differences in muscle tone, or lifestyle influences such as behaviour, diet, or lip hydration. This study also identified that in all generated wireframe diagrams (Figs 2 to 6) the peaks of the Cupid's bow were closely related to the margins of the maxillary central-lateral incisor junctions, with an increase in age presenting an increase in proximity.

## Supporting information

**S1 File.**
(TXT)

**S2 File.**
(TXT)

## Acknowledgments

We are grateful to Dr. Andre Uys for facilitating patient data access, and to the anonymous patients who made this study possible. We would also like to extend our thanks to the reviewers and their contributions, which supported the development of this article.

## Author Contributions

**Conceptualization:** Tobias M. R. Houlton.

**Data curation:** Jason Hemingway.

**Formal analysis:** Tobias M. R. Houlton, Nicolene Jooste, Jason Hemingway.

**Investigation:** Tobias M. R. Houlton, Nicolene Jooste.

**Methodology:** Tobias M. R. Houlton.

**Project administration:** Tobias M. R. Houlton.

**Supervision:** Jason Hemingway.

**Validation:** Tobias M. R. Houlton, Maryna Steyn.

**Visualization:** Tobias M. R. Houlton.

**Writing – original draft:** Tobias M. R. Houlton.

**Writing – review & editing:** Nicolene Jooste, Maryna Steyn, Jason Hemingway.

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
