## [Decision Letter · Decision Letter 0]

14 Jun 2022

PONE-D-22-12251New mathematical approach to mouth approximation: a geometric morphometrics studyPLOS ONE

Dear Dr. Houlton,

Thank you for submitting your manuscript to PLOS ONE. After careful consideration, we feel that it has merit but does not fully meet PLOS ONE’s publication criteria as it currently stands. Therefore, we invite you to submit a revised version of the manuscript that addresses the points raised during the review process.

The reviews are favorable, but include a number of questions and suggestions that should be considered in your revision. One of these relates to how you categorized your subjects.

We look forward to receiving your revised manuscript.

Kind regards,

John Leicester Williams, Ph.D.

Academic Editor

PLOS ONE

Journal Requirements:

Reviewers' comments:

Reviewer's Responses to Questions

**Comments to the Author**

1. Is the manuscript technically sound, and do the data support the conclusions?

Reviewer #1: Yes

Reviewer #2: Yes

2. Has the statistical analysis been performed appropriately and rigorously? 

Reviewer #1: Yes

Reviewer #2: Yes

3. Have the authors made all data underlying the findings in their manuscript fully available?

Reviewer #1: No

Reviewer #2: No

4. Is the manuscript presented in an intelligible fashion and written in standard English?

Reviewer #1: Yes

Reviewer #2: Yes

5. Review Comments to the Author

Reviewer #1: The authors have not fully explained how 'racial origin' was established for the subjects. If the data is split by black/white ancestry then we need to know how this was achieved - was it self-classification by the subjects or assessed by the researcher? Either way, the reliability and ethical basis of such racial classification needs to be established and the data made available if researcher assessment (rather than self-classification) was utilised. Additional comments below:

Abstract

Line 7 – should read ‘Frankfurt/Frankfort Horizontal Plane’

Method

Please be consistent with numbers in the text – if less than 100 please use words (except for % or measurements when numbers are used), if 100 or above use numbers.

Explain how the FHP was utilised in 3D – FHP is a 2D plane only (as the left and right planes may not be parallel due to asymmetry) so how was this mitigated?

Page 8 line 2 – replace ‘done’ with ‘carried out’

Overall

This study is detailed and thorough and provides good data in relation to mouth prediction from skeletal assessment.

However, this study does not provide any practical guide for practitioners as a result of the study (a common problem with geometric morphometric analysis), and this should be addressed before acceptance for publication.

Reviewer #2: First, I will like to salute the effort of the authors in investigating in this domain. But I will also like to comment on few things I think necessary for amendment in your study.

1. Your title (New mathematical approach to mouth approximation: a geometric morphometrics study) needs to be corrected, since it does not align with your study. You don't present any new mathematical formula or any new methodology to the investigation of the soft and hard tissues in this domain.

2. The statement "No existing geometric morphometric studies, however, has considered the spatial relationship between hard and soft facial tissues." needs to be corrected in your introduction section. That was too general; you could have said in South African population. Have you considered these research articles?

(a) The Relationship Between Hard Tissue and Soft Tissue Dimensions of the Nose in Children: A 3D Cone Beam Computed Tomography Study (Eman Allam B.D.S., Ph.D.,Philani Mpofu B.A.,Ahmed Ghoneima B.D.S., Ph.D., M.S.,Mihran Tuceryan Ph.D.,Katherine Kula M.S., D.M.D., M.S) 2018

(b) Soft- and hard-tissue facial anthropometry in three dimensions: what’s new (Chiarella Sforza, Marcio de Menezes* & Virgilio F. Ferrario) 2013

3. What method did you use to calculate the inter-landmark distances?

4. Where is your measurement error table? Can you please display the digitisation error table of the Procrustes ANOVA? Also, which literature do you cite in corroboration with your inter-observer measuring technique or how did you arrive at this method?

5. Which version of MorphoJ and PAST did you use; and where is the citation for the PAST software?

6. What was the allometry effect on the cohorts?

7. The two images in Figure 1 (Visual reference to hard and soft tissue landmark coordinates collected for analysis) should be labelled accordingly such as (a) and (b)

8. If you mention that PC1-PC4 captured the major shape variation, where are the labels in your PCA scatter plot? You need to properly label each of the PC images accordingly in the Figures such as PC1, PC2,.... This will help the reviewers to judge which region(s) contributed to the shape variation in each PC.

9. The arrangement of your conclusion was inter-mixed with your discussion section. I think your conclusion should be transferred to the last paragraph of your discussion section while the last two paragraphs (Starting from: "From this study, we have largely identified shape, size and proportional changes related to demographic variables such as age, population, and sex...") should be transferred to the conclusion section. Note that there is a difference between a conclusion and a summary.

Hope you could look critically into these mentioned observations. Thank you.

6. PLOS authors have the option to publish the peer review history of their article (what does this mean?). If published, this will include your full peer review and any attached files.

Reviewer #1: No

Reviewer #2: **Yes: **Azree Nazri

---

## [Author Response · Author response to Decision Letter 0]

19 Aug 2022

Dear Reviewers,

We are most grateful for your positive feedback and invaluable advice. We have responded to each of your helpful comments and hope the article meets expectations for publication. Details on changes made are listed below. 

1. Manuscript was updated to meet PLOS ONE's formatting requirements (e.g., updates to font size of headings).

2. Regarding participant consent, it is clarified that written consent was obtained by the institution.

3. The compiled raw data has been uploaded as Supporting Information files. 

4. The independent ethics statement has been removed, with all comments on ethics maintained under Materials and Methods.

5. References checked to ensure they are complete and correct.

Comments to Author

Reviewer #1:

1. Population origin was self-prescribed – this is now noted in the second paragraph the Materials and Methods.

2. Text updated to state ‘Frankfurt/Frankfort Horizontal Plane’ where it is stated under Abstract and Materials and Methods.

3. Numbers less than 100 are now written as words (unless they are percentage or measurement values).

4. Under Materials and Methods a description to how the Frankfurt/Frankfort Horizontal Plane was achieved is given (see Pg. 6). 

5. Page 8 line 2 – replaced ‘done’ with ‘carried out’.

6. The wireframes present a visual reference to how the lips tend to be positioned over the dentition. This visual information helps substantiate previous observations used in mouth approximation, and provides a visual guide that indicates how linear measurements are to be placed over the skull. The title has been adjusted to focus the paper as identifying trends in dentition to lip mouth morphology using geometric morphometrics.

Reviewer #2:

1. Title has been modified as recommended.

2. The statement "No existing geometric morphometric studies, however, has considered the spatial relationship between hard and soft facial tissues" has been updated to specifically state that this is in the instance of the South African population.

3. Inter-landmark distances were calculated using Pythagorean Theorem, see page 8 line 17 of paragraph.

4. Measurement error table is available in Table 3, Pg 10. Literature supporting the inter-observer measuring technique is provided in the Materials and Methods, on the last line of Pg 6. 

5. Morpho J v.2.0; PAST v.4.03 – citation included (see Ref [29]).

6. Allometry is something we envisage exploring in more detail in a future study.

7. Figure 1 includes labels (a) and (b), and caption modified accordingly.

8. Labels PC1-PC4 have been included in the scatter plot. 

9. Arrangement of discussion and conclusion have been updated as recommended. Conclusion has been transferred to the last paragraph of the discussion section while the last two paragraphs (Starting from: "From this study, we have largely identified shape, size and proportional changes related to demographic variables such as age, population, and sex...") have been transferred to the conclusion section. 

Many thanks and kindest regards,

Tobias Houlton, 

Nicolene Jooste, 

Maryna Steyn, 

Jason Hemingway

---

## [Editor Report · Decision Letter 1]

23 Aug 2022

Visualising trends in dentition to lip mouth morphology using geometric morphometrics

PONE-D-22-12251R1

Dear Dr. Houlton,

We’re pleased to inform you that your manuscript has been judged scientifically suitable for publication and will be formally accepted for publication once it meets all outstanding technical requirements.

Kind regards,

John Leicester Williams, Ph.D.

Academic Editor

PLOS ONE
---

## [Editor Report · Acceptance letter]

25 Aug 2022

PONE-D-22-12251R1 

Visualising trends in dentition to lip mouth morphology using geometric morphometrics 

Dear Dr. Houlton:

I'm pleased to inform you that your manuscript has been deemed suitable for publication in PLOS ONE. Congratulations! Your manuscript is now with our production department. 

Kind regards, 

on behalf of

Dr. John Leicester Williams 

Academic Editor

PLOS ONE